# Compressive Properties and Energy Absorption Characteristics of Extruded Mg-Al-Ca-Mn Alloy at Various High Strain Rates

**DOI:** 10.3390/ma14010087

**Published:** 2020-12-27

**Authors:** Chongchen Xiang, Zhendong Xiao, Hanlin Ding, Zijian Wang

**Affiliations:** Shagang School of Iron and Steel, Soochow University, 8 Jixue Road, Suzhou 215137, China; ccxiang@suda.edu.cn (C.X.); SudaXiaoZhendong@outlook.com (Z.X.); dinghanlin@suda.edu.cn (H.D.)

**Keywords:** Mg-Al-Ca-Mn alloy, rapid hot extrusion, high strain rate, specific energy absorption

## Abstract

This paper is focused on the mechanical properties and the energy absorption characteristics of the extruded Mg-Al-Ca-Mn alloy in different compression directions under high strain rate compression. Compressive characterization of the alloy was conducted from the high strain rate (HSR) test by using a Split Hopkinson Pressure Bar (SHPB). Results show that the investigated alloy exhibits a strong strain rate sensitivity. With the rise of strain rate, the compressive strength is increased significantly, and the deformation ability also improves. When compressed along the extrusion direction, as the strain rate increases, the total absorbed energy E, the crush force efficiency (CFE), and the specific energy absorption SEA of Mg-Al-Ca-Mn alloy are all greatly improved as compared with those obtained along other compression directions.

## 1. Introduction

The energy and environmental problems brought about by the continuous popularization of automobiles are becoming more and more serious, and they tend to become a key factor hindering the further development of the automobile industry [1]. Light weight components and/or light weight alloys for automobile application have been proved to be an effective way to improve the fuel efficiency and then reduce emissions. Studies show that for every 10% reduction in vehicle weight, CO_2_ emissions can be reduced by 0.09 g/km, which is of great significance for solving energy and environmental problems and maintaining the sustainable development of human society [2,3]. Due to the advantages of light weight, high specific strength, and high specific rigidity, magnesium alloys have received more and more attention and applications in the lightweight of automobiles [4,5].

The energy-absorbing device such as the crash box is an important part of the car bumper system, which plays a role in protecting passengers in the event of a collision and reducing the cost of vehicle repair. At present, the material research of the energy absorption device is mainly concentrated in the field of steel and aluminum alloys [6,7]. Although the studies of the magnesium alloy as an energy-absorbing material are insufficient [8], related studies have also shown that magnesium alloys as automotive energy-absorbing materials have great research value and wide application prospects. Beggs et al. [8] focused on the uniaxial deformation failure mode of AZ31B magnesium alloy pipes that could be used in energy absorption field. Hilditch et al. [9] compared the bearing capacity, energy absorption, and fracture characteristics of three-point bending of deformed magnesium alloy with aluminum alloy pipes and found that AZ31 magnesium alloy exhibits better bearing capacity and energy absorption. These studies have shown that magnesium alloys have great potential in the field of lightweight high-performance energy-absorbing boxes for vehicles. However, these studies are mainly based on commercial AZ31 magnesium alloy, and there is a lack of systematic research that can be related to the microstructure.

Mg-Al-Ca-Mn magnesium alloy has great mechanical properties at high temperature and low price. It is a promising material for applications as a lightweight material [10,11,12,13,14,15,16,17,18]. In addition, due to their homogeneous microstructure and higher mechanical properties, extruded magnesium alloys have attracted special attention than as-cast alloys [18,19,20,21,22]. For example, Homma et al. [23] developed a high-strength extruded Mg-4Al-3Ca-0.3Mn (wt %) alloy with remarkable elevated temperature mechanical properties. 

In this study, a Mg-0.8Al-0.1Ca-0.4Mn magnesium alloy that extruded with a rapid extrusion speed at 30 m/min was developed and taken as the research object, focusing on studying the mechanical properties at high strain rate, and exploring its feasibility as a lightweight material and an automotive energy absorption material. The energy absorption characteristics in different compression directions under high strain rate were also evaluated with the failure mechanism analysis. 

## 2. Materials and Methods 

Mg-Al-Ca-Mn alloy was prepared by using gravity casting with pure Mg, pure Al, Mg-20 wt % Ca and Mg-10 wt % Mn. The nominal composition of Mg-Al-Ca-Mn alloy is established as 0.8 wt % Al, 0.1 wt % Ca, 0.4 wt % Mn, and the rest is Mg by an Inductive Coupled Plasma Emission Spectrometer. The density of this alloy was measured to be 1.74 g/cm^3^.

A 1250-ton horizontal extruder was employed for hot-extruding Mg-Al-Ca-Mn alloy. The size of the extrusion barrel is Φ120 mm, the extrusion barrel temperature and the mold temperature are 450 °C and 430 °C, respectively. The extrusion outlet speed of Mg-Al-Ca-Mn alloy is 30 m/min. The ingot is homogenizing at 400 °C for 8 h before being extruded. The thicknesses of the extruded sheets are 4 mm and 16 mm, respectively. 

Samples for high strain rate compression were designed and prepared as shown in Figure 1. First, round bars with a diameter of 3 mm were turned on a machine tool along the extrusion direction (ED), transverse direction (TD), and horizontal 45° directions separately, which were then processed into Φ3 mm × 3 mm cylindrical samples using a wire cutting machine. We have prepared six samples in each direction for different strain rates.

To further study the mechanical properties of investigated alloy and obtain experimental data over a continuous range of strain rate, a Φ10 mm × 10 mm sample was designed in this study to gain a relatively low strain rate. The schematic diagram of the Φ10 mm × 10 mm sample is illustrated in Figure 2; 16 mm-thick Mg-Al-Ca-Mn alloy was also homogenized at 400°C for 8 h and produced using a 1250-ton horizontal extruder at the same extrusion outlet speed of 30 m/min, which was then cut apart along the middle lines by a wire cutting machine, and the parts with uniform structure in the middle of the plates were taken to prepare samples. Unlike the Φ3 mm × 3 mm samples, the three compression directions selected in this experimental test were ED, normal direction (ND), and the 45° direction of ED turning to ND (hereinafter referred to as “the vertical 45° direction”). We prepared the same number of samples as Φ3 mm × 3 mm cylindrical samples in each direction.

A standard metallography procedure is applied to polish the specimen surface before microscopy, which included sequential polishing down to 1 µm diamond slurry. The Picric etchant containing 5% picric acid, 13% acetic, 70% ethanol, and 12% distilled water is used as the etching solution for Mg-Al-Ca-Mn alloy. A scanning electron microscope (SEM, Hitachi SU5000, Hitachi, Tokyo, Japan) equipped with an Electron Backscatter Diffraction (EBSD) were used for the microstructure analysis. 

A Split Hopkinson Pressure Bar was employed for the high strain rate compression testing. The sample is sandwiched between the incident and transmitter bars. Molybdenum disulfide was applied on the both bottom surfaces as the lubricant. A brass pulse shaper with a diameter of 5 mm was used during the test. Detailed information of the experiment setup can be found from the author’s previous work [24]. Three samples in the same direction were used for the same strain rate in order to get more accurate data. 

It should be noted that the Φ10 mm × 10 mm samples were not compressed into flakes macroscopically, and the macroscopical deformation was not caused by a single impact of the Hopkinson pressure bar; instead, it was the result of multiple impacts in one experimental test. In this work, the mechanical properties and energy absorption performance at different strain rates were all analyzed according to the stress–strain data of the first-pass compressive deformation extracted from the multiple impacts.

## 3. Results

Figure 3 shows the microstructure and pole figure of the as-extruded Mg-Al-Ca-Mn alloy in the thickness of 16 mm. It can be seen from the figure that the microstructure of the Mg-Al-Ca-Mn alloy consists of relatively uniform equiaxed grains, and there is no other structure. The pole figure shows weak texture, low pole density, and an obvious deflection in the Mg-Al-Ca-Mn alloy.

### 3.1. Mechanical Properties of Mg-Al-Ca-Mn Alloy

The stress–strain curves of Mg-Al-Ca-Mn alloy under compression at the strain rates of about 3200/s and about 5500/s are shown in Figure 4, and the corresponding mechanical property parameters are given in Table 1. The Mg-Al-Ca-Mn alloy shows a strong strain rate sensitivity when it is compressed in the ED direction (Figure 4a). When the strain rate increases from about 3200/s to about 5500/s, the yield strength increases from 178.48 to 196.44 MPa, and the compressive strength increases from 572.40 to 739.42 MPa, increasing by about 30%. The total compressive strain related to the first compressive deformation also increases from 10.18% to 15.56%.

Note that when the samples are compressed along the TD (Figure 4b) or the horizontal 45° direction (Figure 4c), the change of the stress with increasing strain under the two strain rates are nearly coincident with each other, indicating a strain rate insensitive as compared to those samples compressed along ED. The corresponding values of yield compressive strength, compressive strength, and total compressive strain are all summarized in Table 1.

By comparing the stress–strain curves of the three compression directions, it can be found that when compressed in the ED direction, the material shows the largest yield strength and compressive strength, a strong strain rate sensitivity, and the least maximum deformation, although the deformation degree changes the most significantly with the increase of strain rate; when compressed in the horizontal 45° direction, the maximum deformation of the material is the largest, but the deformation degree changes the least with the increase of strain rate; and when compressed in the TD direction, the strength of the material is similar to that when compressed in the horizontal 45° direction, but the maximum deformation is lower than that when compressed in the horizontal 45° direction, and the maximum deformation increases significantly with the increase of strain rate.

The difference of mechanical properties shown in the stress–strain curves is essentially the difference of deformation mechanisms, which is related to the deformation of different samples’ direction [25]. The mechanical properties of Mg-Al-Ca-Mn alloy when compressed in the ED direction are obviously different from those in the other two directions, indicating that the mechanism leading to the material deforming in the ED direction is different from that in the other two directions.

With Φ10 mm × 10 mm samples, the mechanical behavior of the alloy under lower strain rates can be acquired with the same experimental setup. The stress–strain curves of Mg-Al-Ca-Mn alloy under high strain rate compression at the strain rate about 700/s and about 1400/s are shown in Figure 5, and the corresponding mechanical property parameters are given in Table 2.

According to Figure 5, the Mg-Al-Ca-Mn alloy shows strong strain rate sensitivity when it is compressed in the ED direction. When the strain rate rises from 700/s to 1400/s, the yield strength grows from 87.90 to 94.37 MPa, and the compressive strength increases from 113.42 to 231.59 MPa. The deformation greatly increases from 3.01% to 5.65% with strain rate, and the hardening rate in the elastic stage also increases. The material is subject to secondary hardening after yielding. When the strain rate reaches 1400/s, the material undergoes secondary hardening, of which the hardening rate is apparently higher than that when the strain rate is 700/s. Then, the material turns soft obviously before it hardens again, of which the hardening rate is generally the same as that of the secondary hardening.

The strain rate sensitivity is relatively weak when compressed in the ND direction, and the curves almost coincide with each other before material yielding, without any marked change for the yield strength. The material is subject to secondary hardening after yielding. The hardening rates are approximately the same under the two strain rates. When the strain rate is 1400/s, the compressive strength and the deformation are significantly improved compared with those when the strain rate is 700/s.

The samples compressed in the vertical 45° direction also manifest low strain rate sensitivity. The yield strength does not obviously increase with the rise of strain rate, and the two stress–strain curves almost coincide with each other before yielding. The secondary hardening rates after yielding are close to each other, but the secondary hardening at the strain rate of 1400/s lags behind that at the strain rate of 700/s. The compressive strength and the deformation increase sharply with strain rate.

Distinct progressive variation can be found for the mechanical properties of Mg-Al-Ca-Mn alloy when compressed in the three directions of ED, vertical 45°, and ND. The largest values of yield strength, compressive strength, and deformation appear when the material is compressed in the ED direction, which is followed by the compression in the vertical 45° direction, and the smallest values appear when compressed in the ND direction. This indicates that the dominant compressive deformation mechanism transits from one to another when the load direction rotates 90° from ED to ND.

Figure 6 collects the stress–strain curves of Mg-Al-Ca-Mn alloy obtained in this study when the material is compressed in the ED direction at the strain rate of 700/s, 1400/s, 3200/s, and 5500/s separately. It can be seen from the figure that the Mg-Al-Ca-Mn alloy displays a strong strain rate sensitivity with the rise of strain rate when it is compressed in the ED direction and the deformation also increases significantly with the rise of strain rate. According to the pole figures in Figure 3, the texture of Mg-Al-Ca-Mn alloy is weak with obvious deflection observed, which indicates that the grains in the matrix are oriented in random, and there are many grains showing soft orientations favorable for slip. A previous research also evidences that [26] when the slip-dominated deformation occurs in magnesium alloy, the material exhibits significant strain rate sensitivity. Therefore, it can be preliminarily determined that the deformation mechanism of Mg-Al-Ca-Mn alloy when compressed in the ED direction is slip-dominated. When the strain rate is more than 3200/s, the stress–strain curves of Mg-Al-Ca-Mn alloy also show softening and secondary hardening after yielding, but the softening is not as obvious as that when the strain rate is 1400/s. A possible reason may be that the dynamic hardening effect is more obvious, although the material undergoes softening when the strain rate increases to a certain value. As a result, the stress–strain curve reflects a decreased slope and a decreased hardening rate, but the strength of the material is still rising.

### 3.2. Energy Absorption Performance of Mg-Al-Ca-Mn Alloy

In a crush resistance test using an automotive crash box, the common parameters for evaluating the energy absorption performance of a material are the peak load P_max_, total absorbed energy E, average crush force P_m_, crush force efficiency CFE, specific energy absorption SEA, and average load P_avr_. These parameters are expanded around the relationship y = P(s) between the crush force P and the deformation length s. Therefore, in order to evaluate the energy absorption performance of the Mg-Al-Ca-Mn alloy in this study, it is necessary to convert the stress–strain curve y = σ(ε) into the crush force–deformation length curve y = P(s).

The calculation of engineering stress and strain is as follows:σ = P/A(1)
ε = (L − L_0_)/L_0_(2)
where P is the load; A is the original cross-sectional area of a sample; L_0_ is the original length of a sample; and L is the length of a deformed sample.

Assuming that the sample volume V keeps unchanged during the deformation, the instantaneous cross-sectional area A(t) = V/(L_0_ − s(t)). Yet, the engineering stress–strain is calculated based on the original cross-sectional area A. Therefore, in order to accurately convert the stress–strain curves into crush force–deformation length curves, the engineering stress–strain curves need to be converted into true stress–strain curves. The conversion equation for its corresponding time t is:σ_true_(t) = σ_eng_(t)∙(1 − ε_true_(t))(3)
ε_true_(t) = −ln(1 − ε_true_(t))(4)
where P(t) = σ_true_(t)∙A(t) and s(t) = L_0_∙ε_true_(t).

The P-s curves in each direction after conversion are shown in Figure 7. The parameters for evaluating the energy absorption performance of the Mg-Al-Ca-Mn alloy when high strain rate is compressed in the three directions can be obtained through calculation. The results are shown in Table 3, Table 4 and Table 5.

It can be seen from Table 3, Table 4 and Table 5 that the total absorbed energy E, the average load P_avr_, and the specific energy absorption SEA of Mg-Al-Ca-Mn alloy are higher when the strain rate increases. The total absorbed energy E reaches 1.98 J, and the specific energy absorption SEA reaches 53.66 J/g, which are the highest values for the Mg-Al-Ca-Mn alloy compressed in the three directions in this study. By comparing it with Figure 4a, it is found that the reason why the total energy absorbed by the Mg-Al-Ca-Mn alloy is rapidly increased with strain rate when compressed in the ED direction is that it has a higher strain rate sensitivity. As the strain rate increases, the compressive strength increases greatly, corresponding to the increase of the crush force P in the P-s curve in Figure 4a. The strain also increases significantly with strain rate, corresponding to the increase of the deformation length s in the P-s curve. In a physical sense, the integral area of the P-s curve is the total absorbed energy E. Therefore, a higher strain rate sensitivity means a substantial increase in the total energy E absorbed by the material.

Since the automotive crash box is generally compressed along the ED direction under service conditions, and the aforementioned Φ3 mm × 3 mm sample compression test has proved that the Mg-Al-Ca-Mn alloy shows the highest energy absorption performance in the ED direction compared to the other test results when testing at a large strain rate, so the following will not discuss the energy absorption performance in the TD and horizontal 45° directions but rather investigate the energy absorption performance in the ED direction and the underlying cause for its change. The P-s curves of Φ10 mm × 10 mm Mg-Al-Ca-Mn alloy samples compressed in the ED direction are shown in Figure 8. Parameters to evaluate the energy absorption performance when the Mg-Al-Ca-Mn alloy is compressed at high speed in the three directions such as the total absorbed energy E, the peak load P_max_, the average crush force Pm, the crush force efficiency CFE, the specific energy absorption SEA, and the average load P_avr_ are calculated. The results are listed in Table 6.

Compared with Table 3, the total absorbed energy of the Φ10 mm 10 mm sample is higher than that of the Φ3 mm × 3 mm sample, and with the rise of strain rate, the crush force efficiency decreases, indicating a significant influence of the sample size on the energy absorption performance. The specific energy absorption SEA is defined as the energy absorbed by the unit weight of material, which is less affected by the material size. It can intuitively reflect how the energy absorption performance of the Mg-Al-Ca-Mn alloy changes under different strain rates. Therefore, the specific energy absorption was employed in this study to compare the sample’s energy absorption performance of the Mg-Al-Ca-Mn alloy. Figure 9 plots the changes of specific energy absorption (SEA) with strain rate when the Mg-Al-Ca-Mn alloy is compressed in the ED direction. It can be seen that when the alloy is compressed under a relevantly lower strain rate, the specific energy absorption rises slowly. However, when the strain rate increases, the specific energy absorption rises exponentially.

Based on the aforementioned study, it can be concluded that when compressed in the ED direction, as the strain rate increases, the total absorbed energy E, the crush force efficiency CFE, and the specific energy absorption SEA of the Mg-Al-Ca-Mn alloy are all greatly improved. The Mg-Al-Ca-Mn alloy shows the highest overall energy absorption performance in this direction compared to the other test results. The reason is that the Mg-Al-Ca-Mn alloy exhibits a higher strain rate sensitivity when compressed in the ED direction. Thus, the Mg-Al-Ca-Mn alloy in this study has a good development potential and a wide application prospect when used as an energy-absorption material for vehicles.

## 4. Discussion

In this section, the authors discuss the results and how they can be interpreted through the perspective of previous studies and the working hypotheses. The findings and their implications are discussed in the broadest context possible. Future research directions are also highlighted.

### 4.1. Microstructure Analysis

Figure 10 is the microstructure of Mg-Al-Ca-Mn alloy compressed in the ED direction. The figure shows that many twins are formed in the coarse primary grains elongated in the ED direction at a strain rate of 700/s. The results manifest that the larger the grain size, the more it is favorable for twins formation [27]. When the strain rate is 1400/s, the twins grow further and the original grain boundary disappears, indicating that the original grains are depleted.

Figure 11 is the microstructure of Mg-Al-Ca-Mn alloy compressed in the ND direction. It can be seen from the figure that when the strain rate is 700/s, the microstructure of the material consists of many twins and a small amount of recrystallized grains. When the strain rate rises to 1400/s, obvious recrystallized grains in the structure have disappeared. For the two strain rates, no obvious adiabatic shear band is found in the material. Generally, the dynamic recrystallization does not occur because the material cannot reach the recrystallization temperature when deformed at room temperature. However, during high strain rate compression, local overheating will occur in the material, resulting in the formation of adiabatic shear bands. The internal stress of adiabatic shear bands is concentrated, and the temperature can reach the recrystallization temperature, triggering dynamic recrystallization. Nevertheless, there is no obvious adiabatic shear band found at both strain rates, indicating that the recrystallized grains observed at the strain rate of 700/s are the recrystallized grains formed during the extrusion of original plates rather than being formed in the dynamic compression test.

Figure 12 is the microstructure of Mg-Al-Ca-Mn alloy compressed in the vertical 45° direction. It can be observed that when the material is compressed in this direction, not only is the structure similar to the formation of a large number of twins in the elongated coarse original grains when compressed in the ED direction, but also the structure resembles the coexistence of coarse twins and original recrystallized grains when compressed in the ND direction in the material. As mentioned above, distinct progressive variation can be found for the mechanical properties of Mg-Al-Ca-Mn alloy when compressed in the three directions of ED, vertical 45°, and ND, which may root in the progressive variation in the microstructure.

### 4.2. Fracture Mechanism of Mg-Al-Ca-Mn Alloy

As shown in Figure 13, the sample compressed in the ND direction is fractured along two small surfaces (one left and one right, both of which are approximately 45° to the loading direction), while the sample compressed in the vertical 45° direction is fractured along a single large surface, which is approximately 45° to the loading direction. It should be noted that the actual fracture surfaces are dependent on the sampling directions of the two samples. In this study, when observing the micromorphology of fractured samples, the observed surface is parallel to the fracture surface. According to Figure 14, for the sample compressed in the vertical 45° direction, the fracture surface is parallel to the elongated coarse original grains, i.e., it is parallel to the ED direction. However, for the sample compressed in the ND direction, no elongated coarse original grains are found on the observed surface, indicating that the observed surface is perpendicular to the ED direction in this case. Consequently, the fracture surface of the sample compressed in the ND direction is about 45° deflecting from that of the sample compressed in the vertical 45° direction.

Figure 15 shows the SEM images of the fracture of Mg-Al-Ca-Mn alloy compressed in the ND direction. The images reveals that the fracture on the sample compressed in the ND direction has two morphology features, one is the stepped and river pattern in Figure 15a,b, while the other is the smooth plane in Figure 15c,d. Nearly without any dimple found on the fracture surface, it is a brittle fracture. The river pattern and the cleavage step are the typical features of cleavage fracture, whereas the smooth plane may be the cleavage surface of elongated coarse original grains in the sample, so the fracture mode when compressed in the ND direction is a transgranular cleavage fracture.

Figure 16 shows the SEM images of the fracture of Mg-Al-Ca-Mn alloy compressed in the vertical 45° direction. It can be seen from the figure that the river pattern and cleavage steps are also observed on the fracture surface when compressed in the vertical 45° direction, but the fracture is smoother as a whole than that formed by the compression in the ND direction. The area with big steps shown in Figure 15a,b is not found on this fracture. This difference is also distinct in Figure 14. Since the fracture is relatively smooth and parallel to the ED direction (i.e., parallel to the crystal face of elongated coarse original grains), the fracture mode may be an intergranular fracture where a crack propagates along the grain boundaries of the elongated coarse original grains.

## 5. Conclusions

In this study, the extruded Mg-Al-Ca-Mn alloys with two sizes (Φ3 × 3 and Φ10 × 10) were compressed in different directions. The energy absorption performance and mechanical properties were studied under different strain rates. The feasibility of using Mg-Al-Ca-Mn alloy as an energy absorption material for vehicles was also analyzed, and the main conclusions are as follows:(1)The extruded Mg-Al-Ca-Mn alloys compressed in the ED direction show the best energy absorption performance. The crush force efficiency CFE and the specific energy absorption SEA of Mg-Al-Ca-Mn alloy were all greatly improved by increasing the strain rate. Therefore, the positive effect can be obtained using the ED direction of extruded Mg-Al-Ca-Mn toward the direction of impact to manufacturing crash box, the damage caused by the frontal impact will be minimized, and the safety of passengers will be protected in the vehicle under the crash.(2)The reason why the specific energy absorption of Mg-Al-Ca-Mn alloy turns much higher is that the Mg-Al-Ca-Mn alloys in the ED direction had a very high strain rate sensitivity, which results in the higher specific energy absorption. Therefore, the compressive strength is significantly improved with the increasing of the strain rate.(3)The non-completely recrystallizing of grains in an extruded Mg-Al-Ca-Mn alloy plate would cause the elongated coarse original grains to remain in the base material. Then, the brittle fracture will occur when the alloys were compressed in the ND or vertical 45° direction under a high strain rate. The fracture mode of the samples compressed in the ND direction was the transgranular cleavage fracture, while the fracture mode of the samples compressed in the vertical 45° direction was the intergranular fracture. The crack propagated along the grain boundaries of the coarse original grains when the samples were compressed in the vertical 45° direction.

## Figures and Tables

**Figure 1 materials-14-00087-f001:**
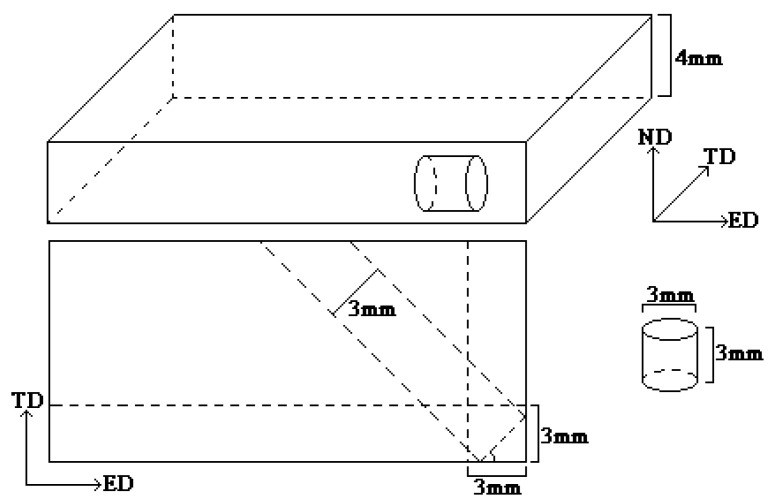
Schematic diagram of preparing Φ3 mm × 3 mm cylindrical samples for high strain rate compression.

**Figure 2 materials-14-00087-f002:**
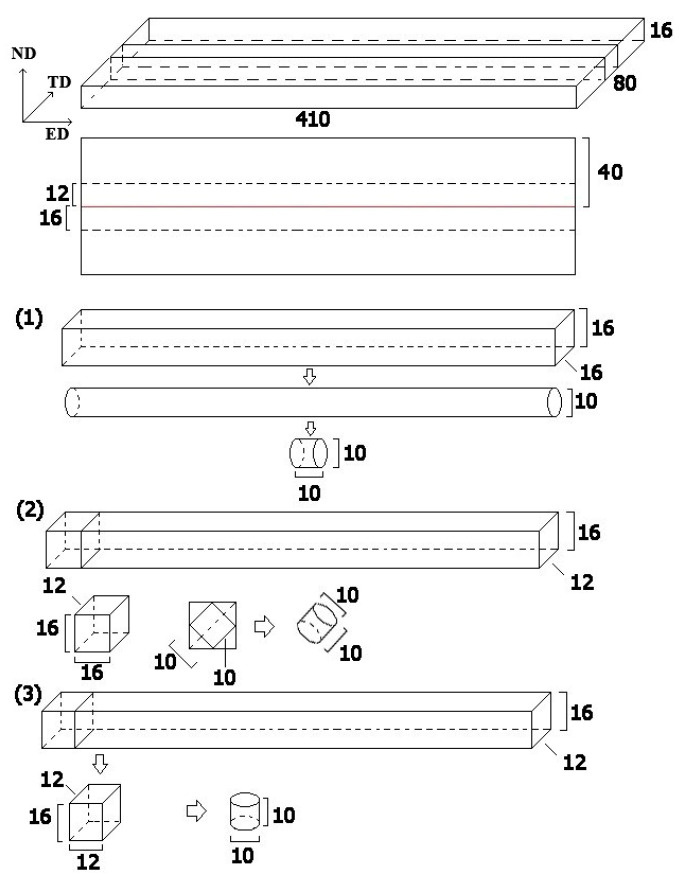
Schematic diagram of preparing Φ10 mm × 10 mm cylindrical samples for high strain rate compression of (1) ED direction samples, (2) Horizontal 45° samples and (3) ND direction samples, all units are in mm.

**Figure 3 materials-14-00087-f003:**
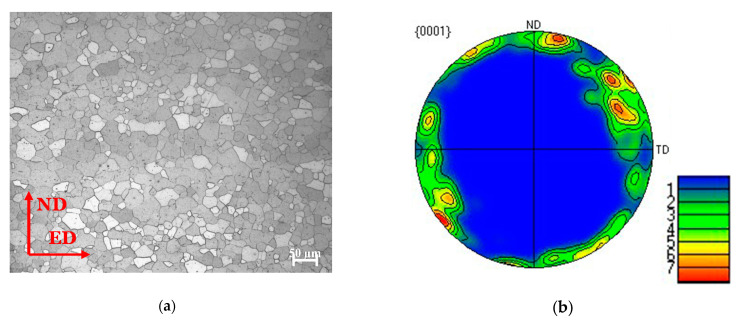
(**a**) The microstructure and (**b**) the pole figure of the as-extruded Mg-Al-Ca-Mn alloy.

**Figure 4 materials-14-00087-f004:**
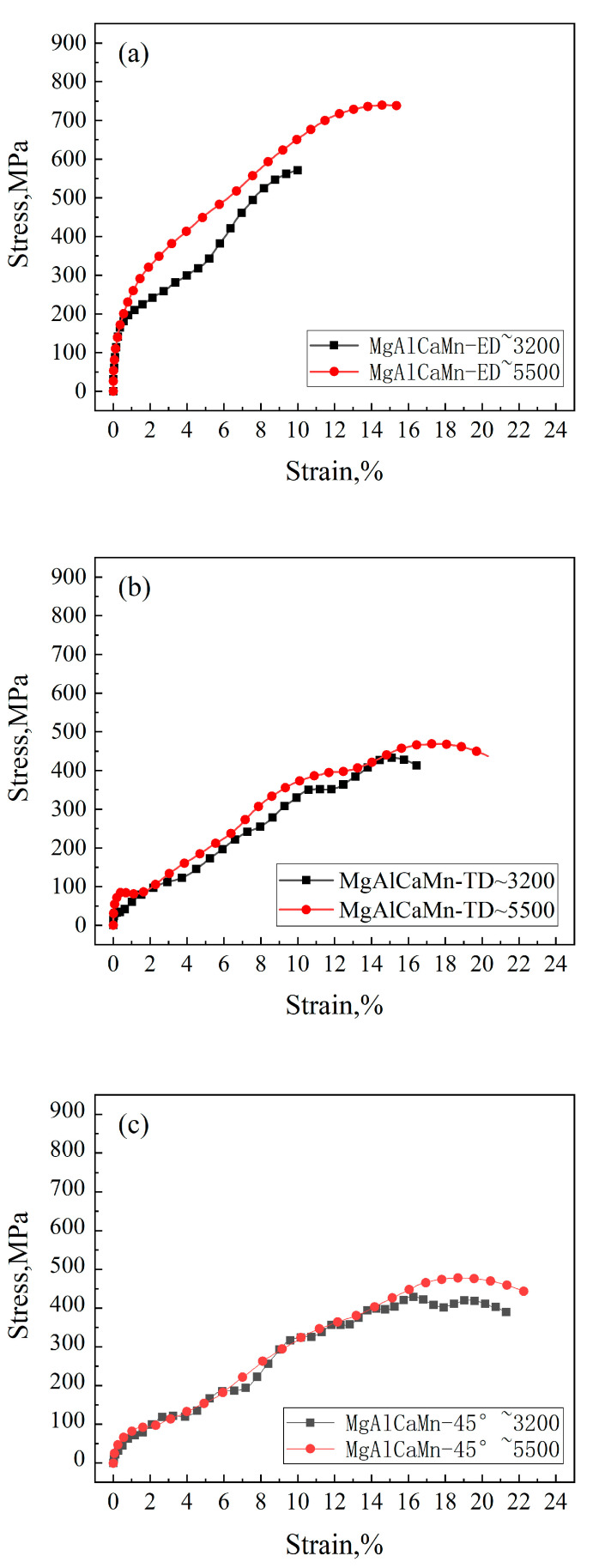
Stress–strain curves of Mg-Al-Ca-Mn alloy under high strain rate compression in the (**a**) extrusion direction (ED), (**b**) transverse direction (TD), and (**c**) Horizontal 45°.

**Figure 5 materials-14-00087-f005:**
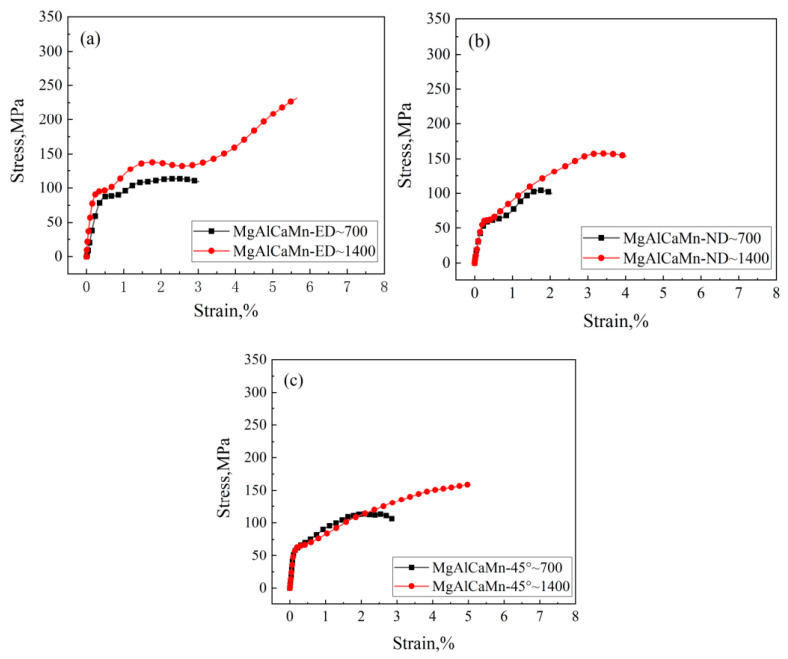
Stress–strain curves of Mg-Al-Ca-Mn alloy under high strain rate compression in the (**a**) ED direction, (**b**) ND direction, and (**c**) Horizontal 45°.

**Figure 6 materials-14-00087-f006:**
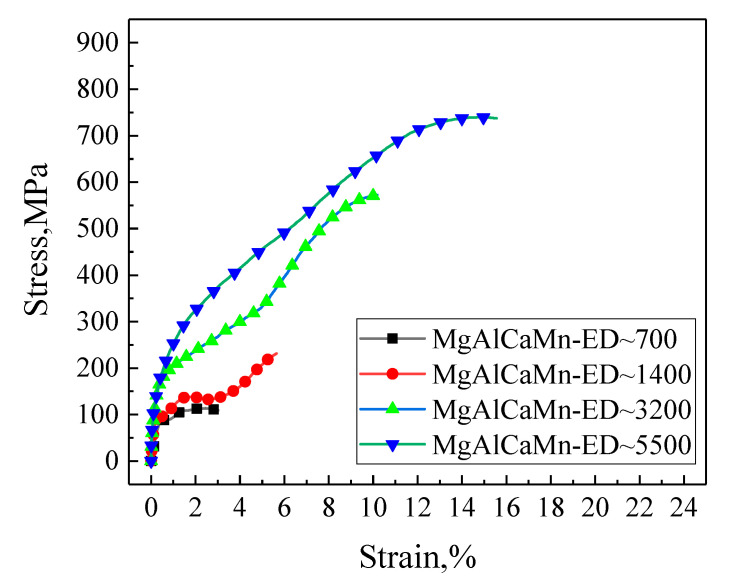
Stress–strain curve of Mg-Al-Ca-Mn alloy dynamically compressed in the ED direction.

**Figure 7 materials-14-00087-f007:**
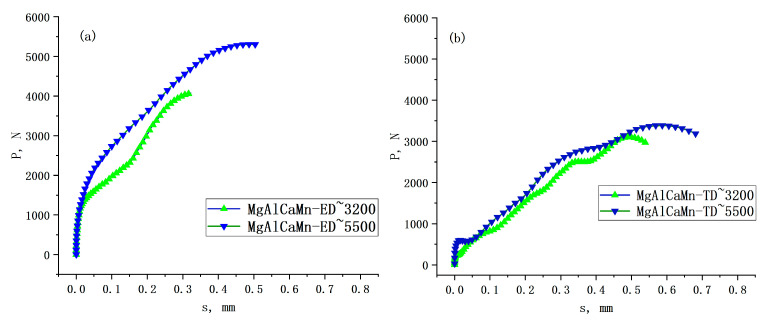
P-s curves of Mg-Al-Ca-Mn alloy under high strain rate compression in the (**a**) ED direction, (**b**) TD direction, and (**c**) Horizontal 45°.

**Figure 8 materials-14-00087-f008:**
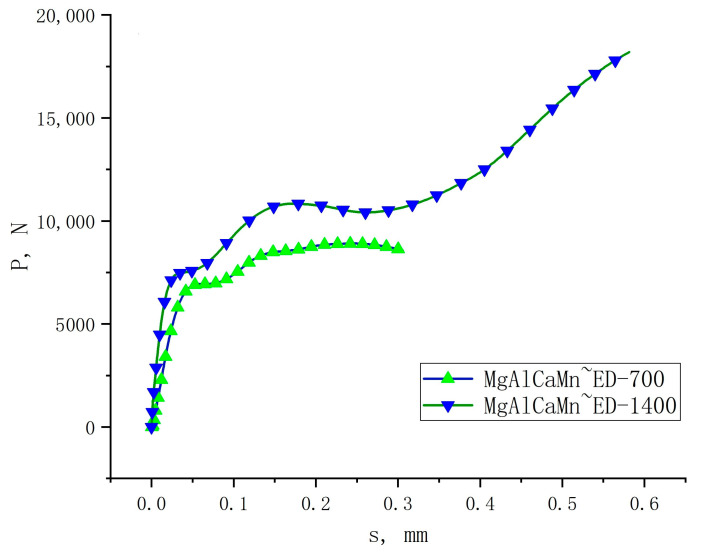
P-s curves of the Mg-Al-Ca-Mn alloy under high strain rate compression in the ED direction.

**Figure 9 materials-14-00087-f009:**
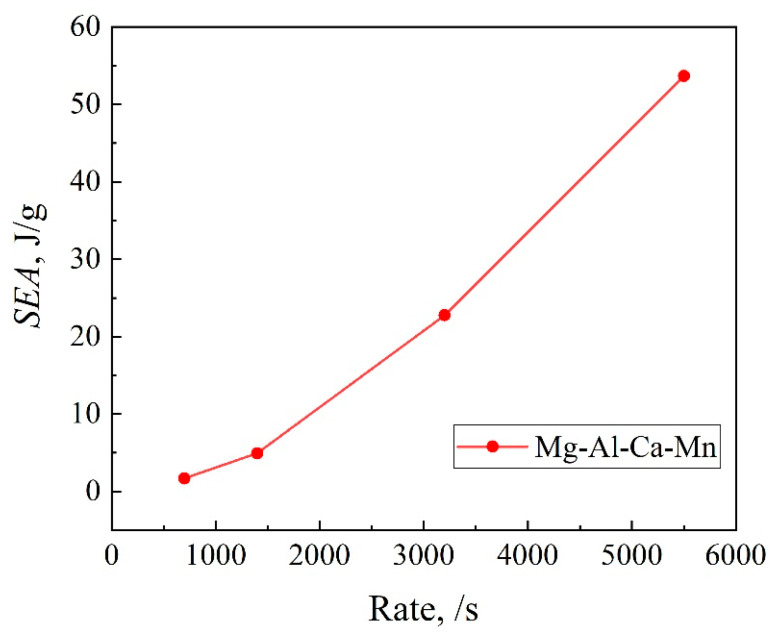
SEA of Mg-Al-Ca-Mn alloy under high strain rate compression in the ED direction.

**Figure 10 materials-14-00087-f010:**
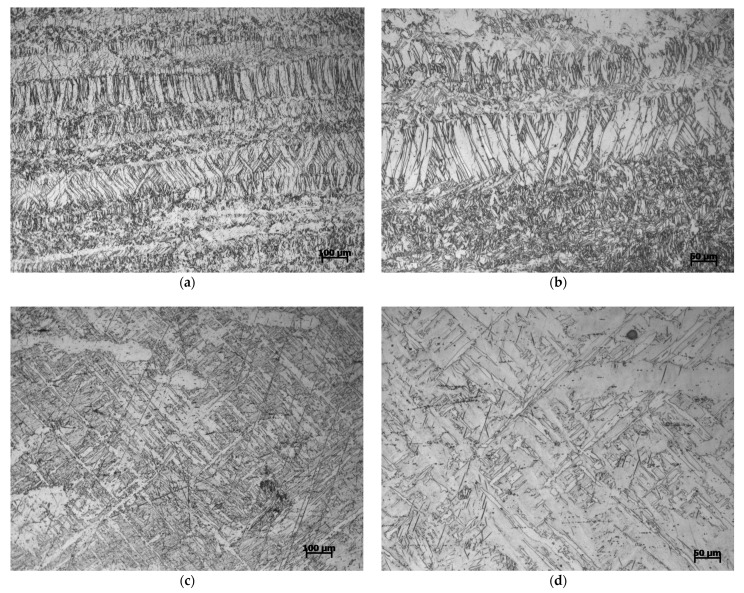
Microstructure of Mg-Al-Ca-Mn alloy compressed in the ED direction under (**a**,**b**) 700/s strain rate and (**c**,**d**) 1400/s strain rate.

**Figure 11 materials-14-00087-f011:**
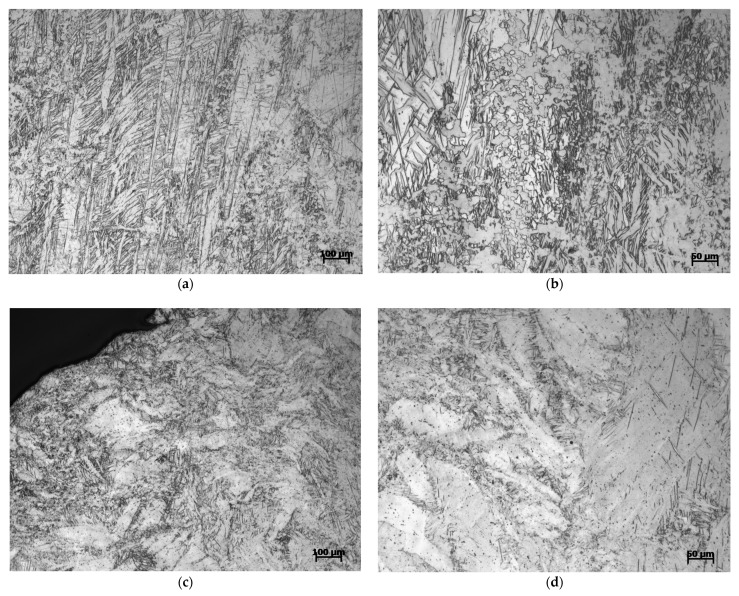
Microstructure of Mg-Al-Ca-Mn alloy compressed in the ND direction under (**a**,**b**) 700/s strain rate and (**c**,**d**) 1400/s strain rate.

**Figure 12 materials-14-00087-f012:**
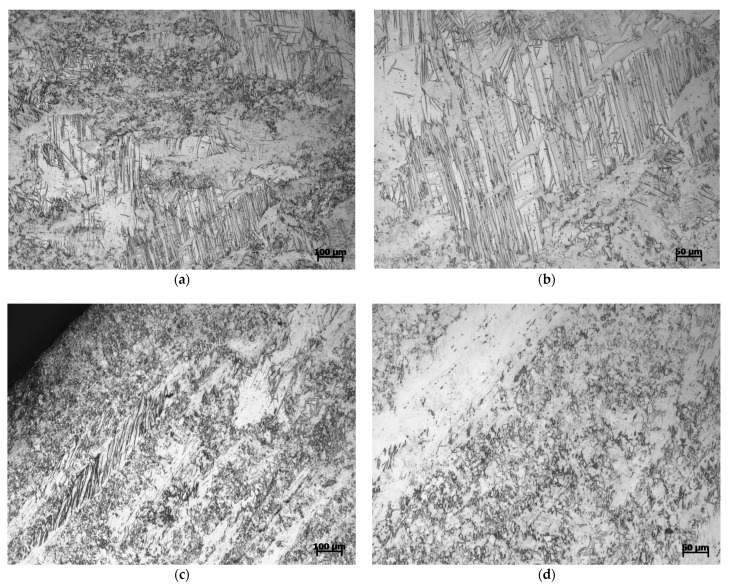
Microstructure of Mg-Al-Ca-Mn alloy compressed in the vertical 45° direction under (**a**,**b**) 700/s strain rate and (**c**,**d**) 1400/s strain rate.

**Figure 13 materials-14-00087-f013:**
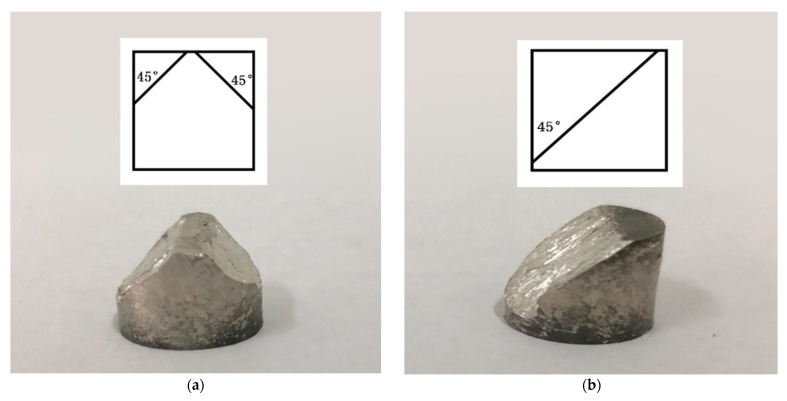
Macroscopic morphology of Mg-Al-Ca-Mn alloy fractured in (**a**) ND and (**b**) vertical 45° direction.

**Figure 14 materials-14-00087-f014:**
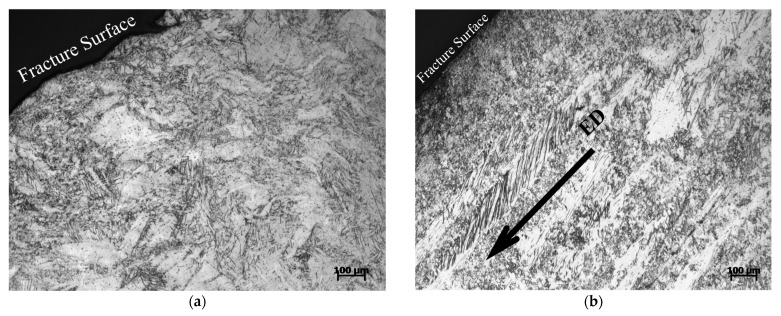
Micro-morphology of Mg-Al-Ca-Mn alloy fractured in (**a**) ND and (**b**) vertical 45° direction.

**Figure 15 materials-14-00087-f015:**
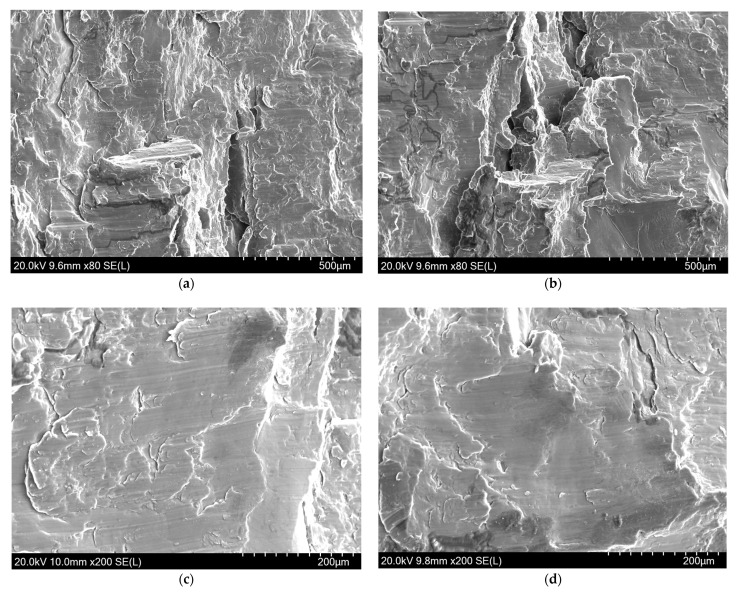
SEM images of Mg-Al-Ca-Mn alloy fractured in the ND direction show (**a**,**b**) the stepped and river patterns and (**c**,**d**) the smooth planes.

**Figure 16 materials-14-00087-f016:**
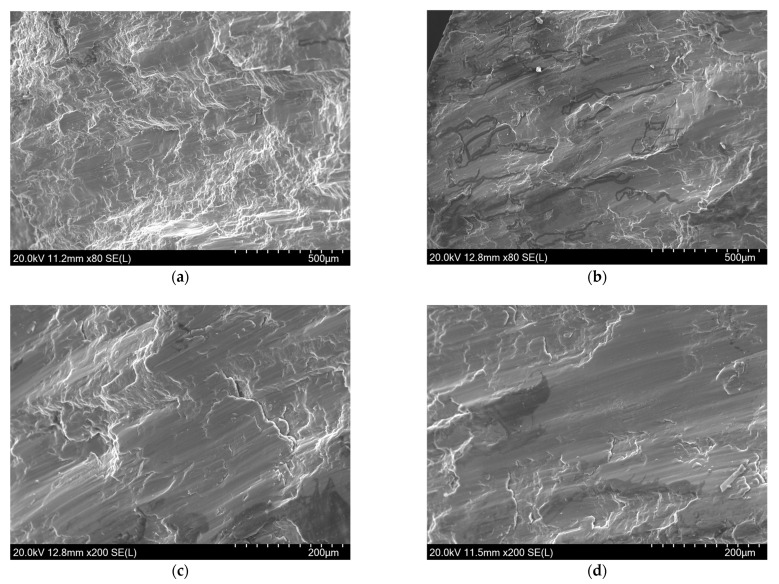
SEM images of Mg-Al-Ca-Mn alloy fractured in vertical 45° direction that show (**a**,**b**) the river patterns without big steps and (**c**,**d**) the smooth planes.

**Table 1 materials-14-00087-t001:** Mechanical properties of Mg-Al-Ca-Mn alloy under high strain rate compression.

Direction	Strain Rate/s^−1^	Yield Stress/MPa	Compressive Stress/MPa	Strain/%
ED	3200	178.48	572.40	10.18
	5500	196.44	739.42	15.56
TD	3200	33.37	432.64	16.61
	5500	73.09	468.73	20.31
Horizontal-45°	3200	31.69	429.71	21.31
	5500	52.61	477.87	22.26

**Table 2 materials-14-00087-t002:** Mechanical properties of Mg-Al-Ca-Mn alloy under high strain rate compression in different directions.

Directions	Strain Rate/s^−1^	Yield Stress/MPa	Compressive Stress/MPa	Strain/%
ED	700	87.90	113.42	3.01
	1400	94.37	231.59	5.65
ND	700	60.20	104.22	2.04
	1400	62.96	157.66	4.01
Horizontal 45°	700	64.77	113.34	2.90
	1400	68.48	174.72	5.04

**Table 3 materials-14-00087-t003:** Energy absorption of Mg-Al-Ca-Mn alloy under high strain rate compression in the ED direction.

	Strain Rate/s^−1^	E/J	P_max_/N	P_m_/N	P_avr_/N	CFE	SEA/(J/g)
Mg-Al-Ca-Mn	3200	0.84	4072.09	2750.49	2042.56	0.68	22.76
	5500	1.98	5304.70	4241.64	2958.10	0.80	53.66

**Table 4 materials-14-00087-t004:** Energy absorption of Mg-Al-Ca-Mn alloy under high strain rate compression in the TD direction.

	Strain Rate/s^−1^	E/J	P_max_/N	P_m_/N	P_avr_/N	CFE	SEA/(J/g)
Mg-Al-Ca-Mn	3200	1.03	3105.58	2067.03	1556.79	0.67	27.91
	5500	1.60	3386.84	2625.96	1975.21	0.78	43.36

**Table 5 materials-14-00087-t005:** Energy absorption of Mg-Al-Ca-Mn alloy under high strain rate compression in the Horizontal 45°.

	Strain Rate/s^−1^	E/J	P_max_/N	P_m_/N	P_avr_/N	CFE	SEA/(J/g)
Mg-Al-Ca-Mn	3200	1.51	3093.50	2361.96	1894.54	0.76	40.92
	5500	1.74	3466.80	2605.57	1996.30	0.75	47.15

**Table 6 materials-14-00087-t006:** Energy absorption of Mg-Al-Ca-Mn alloy under the high strain rate compression in the ED direction.

	Strain Rate/s^−1^	E/J	P_max_/N	P_m_/N	P_avr_/N	CFE	SEA/(J/g)
Mg-Al-Ca-Mn	700	2.33	8908.51	7740.86	6288.46	0.86	1.70
	1400	6.76	18,188.91	11,964.60	9865.19	0.66	4.93

## Data Availability

The data presented in this study are available on request to the corresponding author.

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
