# Peer review of "Compressive Properties and Energy Absorption Characteristics of Extruded Mg-Al-Ca-Mn Alloy at Various High Strain Rates"

_materials, 2020, doi:10.3390/ma14010087_

Round 1
Reviewer 1 Report
Regardless of the overall positive assessment of the reviewed paper, I would like to submit the following comments / suggestions, which in my opinion need to be supplemented / clarified by the Authors:
Materials and Methods:
- The ingot was homogenizing at 400 ℃ for 8 hours before being extruded. Why is such a long homogenization time adopted?
- Was the extrusion outlet speed of the 16 mm product the same as before? Has the same heat treatment been applied? The same parameters as for the description extrusion process of samples 3x3 should be reported.
Results:
Material shows very different sensitivity to both the strain rate and orientation. The relationships between the strain rate, sample orientation and properties are described in detail. However, the stress-strain curves of the Mg-Al-Ca-Mn alloy tested at different strain rates are of a very interesting nature. Therefore the possible reasons of the phenomena identified on their basis (mainly cyclic softening and hardening) should be explained in more detail.
Figures 7,8: The content of the publication presents the meaning of the "P" and "s" values. However, in order to increase the quality of the publication, the axes (force, deformation length) should be accurately described. The scale on the X axis is also unreadable due to the order of magnitude. The unit m is obviously correct, but please consider the description of this axis in millimeters.
3.2 Energy absorption performance of Mg-Al-Ca-Mn alloy:
The authors use the term "two feet" several times and compare these alloys. Only one alloy was tested in the study, the differences result from with different sample dimensions and tests conditions.
Conclusions:
The results contained in the study are very interesting and may be useful for example for improvement the energy absorption performance of structural components. Hence, in the summary of the research results, the authors should consider including short information on how the obtained research results can be applied in practice (suggestion only).
Reviewer 2 Report
The manuscript presents an experimental investigation of the compressive properties and energy absorption characteristics of the extruded Mg-Al-Ca-Mn alloy. In general, the article is well organized and written logically. I recommend to you that this manuscript just needs minor revision. Some minor suggestions must be considered before publication.
- In section 3, lines 102-104. It seems the authors forgot to remove the instructional words from the template. This paragraph should be removed.
- The number of testing samples should be noted in the Materials and Method section.
- The energy absorption property is strongly linked to industrial applications. The application field of the extruded Mg-Al-Ca-Mn Alloy related to the energy absorption properties should be mentioned in the Introduction section.
Reviewer 3 Report
1. This paper focuses on Mg-Al-Ca-Mn alloy, why Zn and Cu are using as raw materials in materials and methods part?
2. In Results part, the first paragraph is "This section may be divided by subheading. It should provide a concise and precise......"what's this sentence meaning for?
3. The sample for figure 3 should be mentioned clearly what kind of sample it was taken from and which direction it is?
4. line 141 page 6, the difference of mechanical properties shown in the stress-strain curves is essentially the difference of deformation mechanisms when the material is compressed for deformation in different direction. Please double check to make the sentence more understandable.
5. In table 2, there is a ND results, while in figure 5 it is TD direction. Which one is the right one? And same question to figure 11.
6. The results in figure 6 is from two kinds of alloys taken from 4mm plate and 16mm plate respectively. Regarding their strength or elongation might in different level, whether it is comparable or not to put them together to reveal the sensitivity of the alloy to strain rate. It is better to supply the tensile strength and elongation of the sample taken from 4mm plate and 16mm plate respectively to make it clear what kinds of the tensile properties they have. The discussion of their strain sensitivity basing on the alloy having the similar properties might be more reasonable. And this will make figure 9 more reasonable as well. Or why not taking the samples having 3mm and 10mm in diameter from the 16mm plate? In this case, the results is comparable.
7. The format of references is not uniform, it should be corrected according to the Instructions of this journal.
Round 2
Reviewer 3 Report
The current paper is reasonably modified and might be accepted for publication. Just one more thing, it sounds the format of references is not similar to the requirement of this journal.
Author Response
Thanks very much for the reviewer's kindly reminder. We have modified the reference to a suitable format according to the requirement of the journal.